# Preclinical Evaluation of the Association of the Cyclin-Dependent Kinase 4/6 Inhibitor, Ribociclib, and Cetuximab in Squamous Cell Carcinoma of the Head and Neck

**DOI:** 10.3390/cancers13061251

**Published:** 2021-03-12

**Authors:** Gabrielle van Caloen, Sandra Schmitz, Cédric van Marcke, Xavier Caignet, Antonella Mendola, Sébastien Pyr dit Ruys, Pierre P. Roger, Didier Vertommen, Jean-Pascal Machiels

**Affiliations:** 1Institut de Recherche Clinique et Expérimentale (Pole MIRO), Université Catholique de Louvain (UCLouvain), Avenue Hippocrate 57, 1200 Brussels, Belgium; gabrielle.vancaloen@gmail.com (G.v.C.); sandra.schmitz@uclouvain.be (S.S.); cedric.vanmarcke@uclouvain.be (C.v.M.); xavier.caignet@gmail.com (X.C.); antonella.mendola@uclouvain.be (A.M.); 2Institut Roi Albert II, Department of Medical Oncology, Cliniques Universitaires Saint-Luc, Avenue Hippocrate 10, 1200 Brussels, Belgium; 3Department of Head and Neck Surgery, Cliniques Universitaires Saint-Luc, Avenue Hippocrate 10, 1200 Brussels, Belgium; 4Protein Phosphorylation Unit, de Duve Institute, Université Catholique de Louvain (UCLouvain), Avenue Hippocrate 74, 1200 Brussels, Belgium; sebastien.pyrditruys@uclouvain.be (S.P.d.R.); didier.vertommen@uclouvain.be (D.V.); 5Cancer Research Center (U-CRC), Université Libre de Bruxelles (ULB), 1070 Brussels, Belgium; pierre.roger@ulb.be; 6Institute of Interdisciplinary Research (IRIBHM) and ULB-Cancer Research Center (U-CRC), Université Libre de Bruxelles (ULB), 1070 Brussels, Belgium

**Keywords:** squamous cell carcinoma of the head and neck, CDK4/6, ribociclib, cetuximab, retinoblastoma, resistance

## Abstract

**Simple Summary:**

We previously showed that ribociclib induces cell cycle arrest in some human papillomavirus (HPV)-negative squamous cell carcinomas of the head and neck (SCCHN) models. However, in vivo, ribociclib has only a cytostatic effect, suggesting that its activity needs to be optimized in combination with other treatments. We investigated the activity of ribociclib in combination with cetuximab in several HPV-negative SCCHN patient-derived tumor xenograft (PDTX) models. We found that the combination of cetuximab and ribociclib was not significantly more active than cetuximab monotherapy. In addition, our observations also suggest that the combination of cetuximab with a cyclin-dependent kinase (CDK) 4/6 inhibitor may reduce the activity of the CDK4/6 inhibitor in some cetuximab-resistant models. Our work has significant clinical implications since combinations of anti-epidermal growth factor receptor (EGFR) therapy and CDK4/6 inhibitors are currently being investigated in clinical trials.

**Abstract:**

Epidermal growth factor receptor (EGFR) overexpression is observed in 90% of human papillomavirus (HPV)-negative squamous cell carcinomas of the head and neck (SCCHN). Cell cycle pathway impairments resulting in cyclin-dependent kinase (CDK) 4 and 6 activation, are frequently observed in SCCHN. We investigated the efficacy of ribociclib, a CDK4/6 inhibitor, in combination with cetuximab, a monoclonal antibody targeting the EGFR, in HPV-negative SCCHN patient-derived tumor xenograft (PDTX) models. The combination of cetuximab and ribociclib was not significantly more active than cetuximab monotherapy in all models investigated. In addition, the combination of cetuximab and ribociclib was less active than ribociclib monotherapy in the cetuximab-resistant PDTX models. In these models, a significant downregulation of the retinoblastoma (Rb) protein was observed in cetuximab-treated mice. We also observed Rb downregulation in the SCCHN cell lines chronically exposed and resistant to cetuximab. In addition, Rb downregulation induced interleukin 6 (Il-6) secretion and the Janus kinase family member/signal transducer and activator of transcription (JAK/STAT) pathway activation that might be implicated in the cetuximab resistance of these cell lines. To conclude, cetuximab is not an appropriate partner for ribociclib in cetuximab-resistant SCCHN models. Our work has significant clinical implications since the combination of anti-EGFR therapy with CDK4/6 inhibitors is currently being investigated in clinical trials.

## 1. Introduction

Squamous cell carcinoma of the head and neck (SCCHN) is the seventh most common cancer worldwide, with approximately 700,000 new cases diagnosed per year [1,2,3,4,5]. The majority of SCCHN are due to tobacco use and/or alcohol consumption. The prevalence of SCCHN attributable to human papillomavirus (HPV) is estimated to be around 25–35% [6,7,8,9,10].

Patients with locally advanced disease are usually treated with chemoradiation or surgery followed by (chemo)radiation. However, between 40–50% of these patients will relapse [11,12,13]. The treatment landscape of recurrent and/or metastatic (R/M) SCCHN has recently changed. The first-line palliative treatment used to be a combination of platinum-based chemotherapy with cetuximab, a monoclonal antibody (mAb) targeting the epidermal growth factor receptor (EGFR) [14]. Recently, the Keynote 048 trial showed that pembrolizumab, a mAb targeting programmed cell death 1 (PD1), as monotherapy improved survival over platinum/5-Fluorouracil (5-FU)/cetuximab in SCCHN patients with tumors expressing PD-L1. Platinum/5-FU/pembrolizumab also improved survival compared with platinum/5-FU/cetuximab in all comers [15].

Despite this advance, the median overall survival (OS) for patients with locally R/M SCCHN remains low at around 12 to 15 months, and the treatment goal is mainly palliation [14,16]. Therefore, additional treatment options to improve OS and predictive biomarkers able to enrich the population who will benefit from treatment are needed. In HPV-negative SCCHN, alteration of the tumor suppressor genes Tumor protein 53 (TP53) and Cyclin-dependent kinase inhibitor 2A (CDKN2A) (occurring in 84% and 58% of cases, respectively), and the amplification of the proto-oncogenes Cyclin D1 (CCND1) and Myelocytomatosis viral related oncogene (MYC) (occurring in 31% and 14% of cases, respectively) results in the activation of cyclin-dependent kinase (CDK) 4/6-CylinD complexes leading to cell cycle progression [17,18,19]. We previously showed that ribociclib, a CDK4/6 inhibitor, induces cell cycle arrest in some HPV-negative SCCHN models [20]. However, in vivo, ribociclib has only a cytostatic effect, suggesting that its activity needs to be optimized in association with other treatments, as demonstrated in breast cancer cases where CDK4/6 inhibitors improve outcomes in combination with endocrine therapy [21].

The EGFR is overexpressed in SCCHN and is linked to poor prognosis [22]. Cetuximab in combination with platinum therapy or radiation therapy improves survival in recurrent and locally advanced disease, respectively [14,23]. Cetuximab monotherapy has also demonstrated activity in patients who progress after platinum therapy. In this clinical context, the objective response rates were between 10% and 13%, and the median OS was between 5.2 and 6.1 months [24]. Resistance to EGFR inhibition and cisplatin has been linked to cyclin D1 overexpression [25,26]. EGFR may induce Extracellular signal-regulated kinases 1/2 (ERK1/2)-dependent Cyclin D1 expression [27]. In addition, a study found an inverse correlation between retinoblastoma protein (Rb) inactivation and EGFR expression in HPV-negative patients [28]. The combination of afatinib (a pan-human epidermal receptor (HER) inhibitor) or lapatinib (an EGFR and HER2 tyrosine kinase inhibitor) with a CDK4/6 inhibitor is synergistic in terms of cell viability reduction in HPV-negative cell lines [28]. These data support the investigation of CDK4/6 inhibitors in combination with EGFR or pan-HER inhibitors.

In a SCCHN patient-derived tumor xenograft (PDTX) model, we have previously reported that ribociclib combined with cetuximab induced slight tumor regression that could not be observed with ribociclib or cetuximab monotherapy. However, in that particular model, the size of the tumor size was not significantly different when treated with the combination, compared to cetuximab or ribociclib alone [20]. Here, we investigated the activity of ribociclib in combination with cetuximab in several additional HPV-negative SCCHN PDTX models to further assess the efficacy of this combination. We showed that this combination had a detrimental effect in cetuximab-resistant PDTX models.

## 2. Materials and Methods

### 2.1. Cell Lines and Cell Culture

CAL27 and FaDu cell lines were purchased from the American Type Culture Collection (Manassas, VA, USA) and maintained in culture according to the manufacturer’s instructions. Cell line identities were confirmed every six months by high-resolution short tandem repeat (STR) profiling with the PowerPlex^®^ 16 HS System (Promega, Madison, WI, USA) in the genetic laboratory (GNEX) at Cliniques universitaires Saint Luc (Brussels). Cell lines were tested for mycoplasma contamination with the MycoAlert™ Mycoplasma Detection Kit (Lonza, Bâle, Switzerland) before being used.

### 2.2. Immunoblot Analysis

Cells were lysed on ice for five minutes in a lysis buffer containing Pierce^®^ RIPA buffer with 1% phosphatase inhibitor cocktail and 1% protease inhibitor cocktail (Thermo Fisher Scientific, Waltham, MA, USA). Protein concentration was evaluated using a Pierce™ BCA Protein Assay Kit (Thermo Fisher Scientific, Waltham, MA, USA). The proteins were separated by sodium dodecyl sulfate polyacrylamide gel electrophoresis (SDS-PAGE) electrophoresis on graduated Mini-Protean^®^ TGX™ 4–15% polyacrylamide gradient gels (Bio Rad, Hercules, CA, USA), and transferred to polyvinylidene fluoride (PVDF) membranes (Bio Rad, Hercules, CA, USA). The membranes were blocked with 5% non-fat milk in tris-buffered saline, with Tween^®^ 20, pH 8.0 (Sigma Aldrich, St. Louis, MO, USA) for one hour at room temperature (RT), and subsequently incubated overnight at 4 °C with primary antibodies. After washing, the membranes were then incubated for one hour at RT with horseradish peroxidase (HRP)-conjugated secondary antibodies (BeckmanCoulter, Villepinte, France). The membranes were finally incubated with a HRP substrate to enhance chemiluminescence (ECL) (Thermo Fisher Scientific, Waltham, MA, USA). Chemiluminescence was then revealed with the Spectramax^®^ (GE Healthcare, Diegem, Belgium) or by exposure to CL-Xposure Film (Thermo Fisher Scientific, Waltham, MA, USA) incubated in Kodak developer, followed by Kodak fixer (Sigma Aldrich, St. Louis, MO, USA). Antibodies for immunoblotting were purchased from Sigma Aldrich, St. Louis, MO, USA (GAPDH #G9545), from Cell Signaling Technology, Danvers, MA, USA (Rb #9309, p-Rb (Ser795) #9301, CDK4 #12790, CDK6 #3136, cyclinD1 #2922, CyclinA2 #4656, cyclinE1 #4129, E2F1 #3742, EGFR #4267, pEGFR (Tyr1068) #3777, Akt #9272, p-Akt (Ser473) #4060, c-myc #56605, Erk1/2 #9102, p-Erk1/2 (Thr202/Tyr204) #4370, STAT3 #9139, pSTAT3 (Ser727) #9136, pSTAT3 (Tyr705) #9145, β-catenin #8480), and from Santa Cruz Biotechnology, Dallas, TX, USA (CDK2 #sc-6248, p107 #sc-318, p130 #sc-317, FOXM1 #sc-500).

### 2.3. Establishment of Patient-Derived Tumor Xenograft (PDTX) Mouse Models

PDTX models were derived from three patients with SCCHN, as described previously [20,29]. They were either established in collaboration with Trace, the PDTX platform of KU Leuven, Belgium (HNC002, HNC010), or generated in our laboratory (UCLHN01). The HNC002 model was derived from a patient with hypopharynx cancer that had never been previously treated with systemic therapy, HNC010 was derived from a patient with hypopharynx cancer that progressed after cetuximab and platinum-based therapy, and UCLHN01 was derived from a patient with oral cavity cancer that progressed after chemoradiation. In one model (HNC002-ResCTX), we induced resistance to cetuximab by treating this model weekly with 30 mg/kg of cetuximab. All models were HPV-negative (Appendix A). Mice were randomly divided into four groups and treated with 100 mg/kg ribociclib (daily), 30 mg/kg cetuximab (weekly), a combination of ribociclib and cetuximab (daily/weekly), or vehicles (daily/weekly). Ribociclib was administrated by oral gavage and cetuximab was administrated by intraperitoneal injection. Tumor size was measured by caliper once a week and calculated according to the following equation: V (mm^3^) = [(the largest length) × (the shortest length)^2^]/2. Animal work was undertaken in compliance with Belgian laws and all experiments were in accordance with our local ethics committee (approval number: 2015/13AOU/445). Animal welfare is regularly controlled by inspections under Belgian laws, and all investigators performing animal work had successfully completed the Federation of European Laboratory Animal Science Associations (FELASA) Function C training. The number of mice to be included per group was calculated as previously described [29].

### 2.4. Immunohistochemistry on PDTX Tumor Samples

Immunostaining was performed as previously described by Bouzin and colleagues [30]. Sections were incubated overnight at 4 °C with the primary antibody (Abcam, Cambridge, UK: Rb (#Ab181616, 1/1000); Thermo Scientific, San Jose, CA: Ki67 (#PA1-38032, 1/100)). Stained tissue sections were digitized using an SCN400 slide scanner (Leica Biosystems, Wetzlar, Germany) at 40× magnification. Immunostainings were analyzed using the image analysis tool TissueIA version 4.0.7 (Leica Biosystems, Dublin, Ireland). Rb and Ki67 expressions were determined using a nuclear software algorithm and graded as negative, weak, moderate, or strong. A histoscore with a potential range of 0–300 was calculated as follows: % weakly stained cells + (% moderately stained cells) × 2 + (% strongly stained cells) × 3. Results are presented as mean ± SD of at least three different tumors [31].

### 2.5. Sample Preparation for Mass Spectrometry Analysis

Sample preparation for mass spectrometry analysis was performed as previously described [20].

### 2.6. Label-Free Differential Two Dimension-Liquid Chromatography Coupled to Tandem Mass Spectrometry (2D-LC-MS/MS)

2D-LC-MS/MS analysis was performed as previously described [20].

### 2.7. Small Interfering RNA Transfection

Small interfering RNA (siRNA) targeting retinoblastoma gene (RB1) (#L-003296-02-0005) and negative control siRNA (#D-001810-01-05) were purchased from Dharmacon, CO, Lafayette, USA. Cells were transfected for 72 h with 50 nM siRNA using a Lipofectamine RNAiMAX transfection reagent (Thermo Fisher Scientific, Waltham, MA, USA).

### 2.8. Interleulin-6 Levels in the Cell Culture Supernatant

To measure secreted IL-6, cells were plated in serum-free media or complete media, and an enzyme-linked immunosorbent assay (ELISA) Quantinkine ^®^ ELISA Human Immunoassay (#D6050, R&D Systems, Inc. Minneapolis, MN, USA) was performed on cell culture supernatants collected after 24 h.

### 2.9. Statistical Analysis

Statistical analyses were performed with Prism 8 (GraphPad Software Inc., La Jolla, CA, USA) and R (version 3.5.1, http://www.R-project.com, accessed on 16 November 2019). To compare two independent groups, the unpaired Student’s *t*-test was used, where a *p*-value < 0.05 was considered to be statistically significant. To compare protein expression between cetuximab-resistant and parental cell lines, the paired Student’s *t*-test was used and a *p*-value < 0.05 was considered to be statistically significant. Two-way ANOVA analyses (fixed effects: time and group) with Tukey correction for multiple post-test comparisons were performed to compare the tumor growth of SCCHN xenografts in PDTXs. We used the freely available R/Bioconductor package Pathview [32] to visualize the differential expression of cetuximab-resistant cell proteins compared to their parental lines (CAL27 and FaDu) obtained by proteomic analyses.

## 3. Results

### 3.1. Cetuximab Is Not Always an Appropriate Partner for Ribociclib in PDTX Models

The activity of cetuximab combined with ribociclib was investigated in five HPV-negative SCCHN PDTX models. We specifically chose these PDTX models because they had different sensitivities to cetuximab and ribociclib in monotherapy, allowing us to test the combination in different relevant settings: a cetuximab- and ribociclib-sensitive model (HNC002), two cetuximab-sensitive models with limited or nonactivity of ribociclib (UCLHN01 and HNC007, respectively), and two cetuximab-resistant models in which ribociclib had demonstrated antitumoral activity (HNC010 and HNC002-ResCTX) (Figure 1).

In HNC002, cetuximab or ribociclib treatment induced tumor size stabilization compared to the controls (cetuximab vs. control *p*-value < 0.001; ribociclib vs. control *p*-value: 0.0002). The combination of cetuximab and ribociclib induced slight tumor regression, but the tumor size was not significantly different when treated with the combination, as compared to cetuximab or ribociclib alone (cetuximab vs. combination *p*-value: 0.590; ribociclib vs. combination *p*-value: 0.222) (Figure 1a).

In UCLHN01 and HNC007, cetuximab induced a strong tumor growth delay relative to the controls (UHLHN01: cetuximab vs. control *p*-value < 0.0001; HNC007: cetuximab vs. control *p*-value < 0.0001). Ribociclib induced tumor growth delay compared to the control (ribociclib vs. control *p*-value < 0.0001). HNC007 was resistant to ribociclib, and tumors treated with ribociclib grew faster than in the control group (ribociclib vs. control *p*-value: 0.002). In both models, the combination of cetuximab and ribociclib induced similar tumor size stabilization compared to cetuximab alone (UCLHN01: cetuximab vs. combination *p*-value > 0.999; HNC007: cetuximab vs. combination *p*-value: 0.974) (Figure 1b,c).

The HNC010 model was resistant to cetuximab (cetuximab vs. control *p*-value > 0.999), but ribociclib induced tumor growth delay relative to the controls (ribociclib vs. control *p*-value < 0.0001). Surprisingly, the combination of cetuximab and ribociclib was significantly less active than ribociclib monotherapy (combination vs. ribociclib *p*-value = 0.002) and the tumor growth of cetuximab/ribociclib-treated mice was similar to the controls (combination vs. control *p*-value = 0.142) (Figure 1b). To confirm this observation, HNC002, which was initially sensitive to cetuximab and ribociclib, was chronically treated with cetuximab until resistance ensued, as previously described [29]. This cetuximab-resistant model was named HNC002-ResCTX (Figure 1c). In HNC002-ResCTX, cetuximab had no activity (control vs. cetuximab *p*-value > 0.999), and the Ki-67 protein expression was similar to that of the parental model (Appendix A). As observed in HNC010, the combination of cetuximab and ribociclib was less efficient than ribociclib alone in inducing tumor growth delay relative to the controls (combination vs. control *p*-value = 0.121; ribociclib vs. control *p*-value = 0.0004), although the difference in tumor growth between the ribociclib/cetuximab and ribociclib groups was not statistically significant (combination vs. ribociclib *p*-value = 0.468).

Together, these experiments strongly suggest that the combination of cetuximab and ribociclib is not appropriate in all SCCHN PDTX models.

### 3.2. Cetuximab Induces Downregulation of Retinoblastoma Protein Expression in Cetuximab-Resistant SCCHN PDTX Models Exposed to Cetuximab

To understand how chronic exposure to cetuximab could decrease the efficacy of ribociclib in SCCHN, we used immunohistochemistry to investigate the differences in protein expression profiles between cetuximab-treated tumors and the controls in the PDTX models.

Because retinoblastoma (Rb) are directly regulated by CDK4/6-CyclinD complexes and play an essential role in cell cycle regulation [33], Rb protein expression was investigated in the PDTX models (Figure 2). In the in vivo PDTX experiments described above, mice were sacrificed at the end of the experiment, and tumors of three mice per group were analyzed by immunochemistry. Interestingly, in the two cetuximab-resistant models (HNC010 and HNC002-ResCTX), Rb protein expression measured by histoscore was significantly downregulated in tumors chronically exposed to cetuximab (in combination or not with ribociclib) compared to the controls or the ribociclib monotherapy group (Table 1). However, in the cetuximab-sensitive models (HNC002 and HNC007), there were no differences in the level of Rb protein expression between cetuximab-treated and non-treated tumors (Table 1). In UCLHN01 and HNC007 models, due to the important efficacy of cetuximab and/or combination treatment, some of the tumors were too small to be analyzed for protein expression (Figure 2 and Table 1).

Downregulation of Rb has been linked with decreased ribociclib activity in some preclinical models [20]. Therefore, our data suggest that the downregulation of Rb protein expression after prolonged exposure to cetuximab may explain the decreased efficacy of the ribociclib/cetuximab combination in cetuximab-resistant PDTX models.

### 3.3. Chronic Exposure to Cetuximab Induces Downregulation of the Level of Retinoblastoma Protein Expression in HPV-Negative SCCHN Cell Lines

To investigate the underlying molecular mechanisms, simplified non-heterogenous SCCHN cell cultures were used. CAL27 and FaDu, two cetuximab-sensitive SCCHN cell lines, were chronically treated twice a week with 200 nM cetuximab until resistance occurred.

Cetuximab-resistant cells and their respective parental cell lines were then compared using proteomic analyses. The ratio of the level of protein expression comparing cetuximab-resistant cells to their parental cells was mapped into a network map. Interestingly, the resistant cell lines displayed similar protein profiles, but their protein profiles differed from those of their parental lines. Moreover, as observed in the SCCHN PDTX models, prolonged exposure to cetuximab (three weeks) induced downregulation of the level of Rb protein expression in the two SCCHN cell lines, as compared to the controls (Figure 3 and Figure 4). Surprisingly, despite the negative regulation of Rb, a tumor suppressor protein, many proteins related to cell cycle progression were downregulated following the cetuximab treatment (e.g., CDK4 and cyclin A2) (Figure 3 and Figure 4). No clear increased or reduced activity of the Phosphatidylinositol-3-kinase/protein kinase B (PI3K/Akt) and the Mitogen-activated protein kinases (MAPK) pathways was observed in the cetuximab-resistant lines relative to their parental lines (Figure 3 and Figure 4). On the contrary, the levels of expression of many proteins related to the JAK/STAT pathway were upregulated in resistant lines compared to their parental lines (Figure 3 and Figure 4). Figure 3 and Figure 4 show the expression levels of proteins identified as important regulators of the cell cycle, PI3K/Akt, MAPK, and JAK/STAT pathways. Complete pathway networks are available in Appendix A.

Results obtained by proteomic analyses were confirmed in triplicate by western blotting assays using three different batches of cell lines (Figure 5 and Appendix A). Parental and cetuximab-resistant lines were analyzed as pairs. The Rb downregulation observed in cetuximab-resistant lines compared to their parental lines and the reduced activity of the cell cycle pathways were confirmed (Figure 5a). Moreover, both resistant lines demonstrated increased phosphorylation of STAT3 at tyrosine 705 and serine 727. However, the resistant lines displayed reduced activity of the PI3K/Akt and MAPK pathways relative to their parental lines. Indeed, the phosphorylation of Akt and Erk, as well as the expression of some of their downstream targets such as c-myc or MDM2, were downregulated under prolonged exposure to cetuximab (Figure 5b).

Pre-clinical works have suggested that Interleukin-6 (IL-6) and its downstream signaling proteins such a STAT3 could be implicated in resisting anti-EGFR therapies [34]. IL-6 was increased in the cell culture supernatants of our CAL27 and FaDu cetuximab-resistant cell lines compared to parental cells (Figure 6a). Furthermore, incubation of the CAL27 and FaDu cells with siRNA targeting RB1 increased Il-6 levels in the cell culture supernatants (Figure 6b), supporting the hypothesis that cetuximab-induced Rb downregulation could activate STAT3 through Il-6 secretion.

Taken together, these in vitro data confirmed the downregulation of Rb protein expression induced by prolonged exposure to cetuximab observed in vivo. Further, increased expression of proteins related to the JAK/STAT pathway was detected in cell lines under prolonged exposure to cetuximab.

## 4. Discussion

Our experiments show that cetuximab is not always an ideal partner for ribociclib, particularly in cetuximab-resistant PDTX models.

In the cetuximab-resistant mice, we found that the continuous administration of cetuximab, with or without ribociclib, decreased Rb expression. Rb downregulation was not observed in the cetuximab-sensitive SCCHN PDTX models. Similarly, in our two SCCHN human cell lines with acquired resistance to cetuximab, Rb was also downregulated compared with the cetuximab-sensitive parental cell lines. We previously found a positive correlation between Rb expression levels and the activity of ribociclib in SCCHN preclinical models, providing a possible explanation for the limited antitumoral effect of ribociclib in combination with cetuximab in these cetuximab-resistant models [20]. Similar to ribociclib, palbociclib, and abemaciclib are orally available small molecules that target CDK4/6 which are approved by the Food and Drug Administration and the European Medical Agency. These three inhibitors are structurally similar and interact specifically with residues in the catalytic ATP-binding clefts of CDK4 and CDK6 [35,36,37,38]. Palbociclib, ribociclib and abemaciclib are potent inhibitors of Rb phosphorylation. Several in vitro and in vivo studies have shown that the activity of palbociclib, ribociclib, and abemaciclib correlate with Rb protein expression in various type of tumors [20,39,40,41,42,43,44,45,46,47]. Therefore, we can assume that the downregulation of Rb protein expression induced by cetuximab treatment in cetuximab-resistant tumors might also decrease the activity of palbociclib and abemaciclib.

Rb loss may reflect the transformation of cancer cells to a more aggressive phenotype after continuous exposure to anti-proliferative drugs. Indeed, Niederst and colleagues found that Rb was lost in 100% of tumor samples and cell lines derived from EGFR mutant non-small cell lung cancer patients harboring resistance to anti-EGFR tyrosine kinase inhibitors when small cell lung cancer (SCLC) characteristics are being acquired [48]. In their study, Rb loss was found to be induced by RB1 mutations or deletions. Similarly, in lung adenocarcinomas occurring in non-smokers, RB1 inactivation through complex rearrangements was found in EGFR-mutant tumors and seemed to favor SCLC or squamous cell transformation [49]. We did not find RB1 genomic alterations in our cetuximab-resistant SCCHN PDTX models, but further studies are needed to investigate if Rb loss is associated with cetuximab resistance in SCCHN.

In our cetuximab-sensitive SCCHN PDTX models (HNC002, HNC007, and UCLHN01), we did not find any statistically significant benefits when cetuximab and ribociclib were combined, as compared to cetuximab monotherapy. However, it is possible that the activity of the CDK4/6 inhibitor in combination with cetuximab was blinded by the substantial sensitivity of these models to cetuximab. Several pre-clinical works have suggested a potential benefit of combining anti-EGFR therapy and CDK4/6 inhibitors. Beck and colleagues demonstrated that afatinib or lapatinib combined with palbociclib had promising activity in terms of tumor cell viability inhibition in SCCHN cell lines [28]. Moreover, they found that the combination of lapatinib and palbociclib reduced ERK1/2 phosphorylation in a deeper and more durable way relative to lapatinib as a single agent in SCCHN cell lines. However, in this study, the combination was not investigated in in vivo preclinical models. Wang and colleagues have recently shown a synergistic inhibition of colorectal cancer (CRC) cell growth through simultaneous inhibition of the EGFR and CDK4/6 pathways [50]. Furthermore, the activity of the combination of cetuximab with palbociclib was evaluated in a cetuximab-sensitive CRC PDTX model with wild-type Kirsten rat sarcoma 2 viral oncogene homolog (KRAS)/Neuroblastoma RAS viral oncogene homolog (NRAS)/b-raf proto-oncogene serine/threonine kinase (BRAF) but, similarly to our findings, they did not observe a statistically significant benefit when cetuximab was combined with a CDK4/6 inhibitor, as compared to cetuximab monotherapy [50]. Therefore, additional work is required in terms of cetuximab dosage and sequence to further assess the potential of this treatment combination in cetuximab-sensitive models. In addition, as the CDK4/6-cyclinD complexes act downstream of several oncogenic pathways activated by mechanisms of resistance to cetuximab treatment, future experiments should determine if the addition of CDK4/6 inhibitors could block or postpone acquired resistance to cetuximab in SCCHN.

In our two SCCHN cell lines, cetuximab exposure led to negative regulation of the PI3K/Akt and MAPK pathways, although the activation of these pathways has sometimes been incriminated as a resistance mechanism to EGFR inhibitors in SCCHN [51,52]. Our in vitro data suggest that cetuximab-induced Rb downregulation could activate STAT3 through Il-6 secretion. Previous studies have shown that STAT3 can be activated by both EGFR-dependent and -independent pathways and that STAT3 inhibition can result in tumor growth inhibition and/or apoptosis [53,54,55,56,57]. In correlation with previous observations made in SCCHN preclinical studies [58,59], when CAL27 and FaDu were chronically exposed to cetuximab they displayed increased STAT3 activity relative to their parental lines. In addition, several studies have demonstrated that STAT3 knockdown can overcome resistance to EGFR inhibition in SCCHN cells [59,60,61,62]. Taken together with our findings, this suggests that STAT3 activation might contribute towards the maintenance of cell cycle progression under cetuximab in these cell lines. The cetuximab-resistant HNC010 model harbored a PIK3CA hotspot mutation in exon 9, G1624A (E542K) that may explain this resistance. However, such a mutation was not found in the HNC002-ResCTX model and the two SCCHN cell lines (CAL27 and FaDu), suggesting the implication of other mechanisms that are currently under investigation.

A phase 2 trial investigated the activity of palbociclib and cetuximab in platinum-resistant and cetuximab-resistant HPV-negative SCCHN. The objective response rates were 39% in platinum-resistant patients and 19% in cetuximab-resistant patients, respectively [63]. Despite these promising findings, a recent randomized phase II trial (PALATINUS) failed to demonstrate a statistically significant benefit to palbociclib combined with cetuximab over cetuximab alone in unselected HPV-negative SCCHN patients who progressed after platinum therapy [64]. Our findings may partly explain the negative results observed in the PALATINUS trial.

## 5. Conclusions

We investigated the activity of ribociclib in combination with cetuximab in several HPV-negative SCCHN PDTX models to assess the efficacy of this combination. We showed that this combination has a detrimental effect in some PDTX models. The combination of anti-EGFR therapy and CDK4/6 inhibitors is currently under clinical investigation. Therefore, the development of the association of cetuximab with a CDK4/6 inhibitor in the clinic should be performed cautiously. This strategy should be deeply investigated in relevant preclinical studies to optimize patient selection and to increase the likeliness of improving patient outcomes.

## Figures and Tables

**Figure 1 cancers-13-01251-f001:**
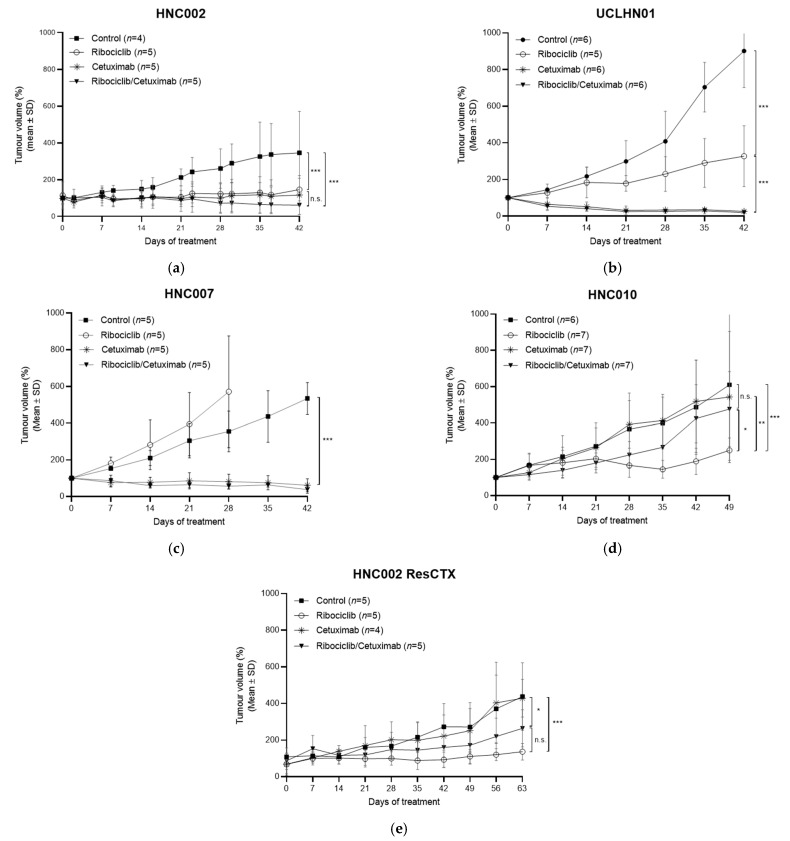
Ribociclib, cetuximab, and ribociclib/cetuximab activity were evaluated in three HPV-negative SCCHN PDTX models; (**a**) HNC002, (**b**) UCLHN01, (**c**) HNCOO7, (**d**), HNC010, (**e**) HNC002-ResCTX. Mice received daily 100 mg/kg ribociclib, or weekly 30 mg/kg cetuximab, a combination of ribociclib and cetuximab, or the vehicles. n.s. (no significance): *p* > 0.05; * *p* < 0.05; ** *p* < 0.001; *** *p* < 0.0001.

**Figure 2 cancers-13-01251-f002:**
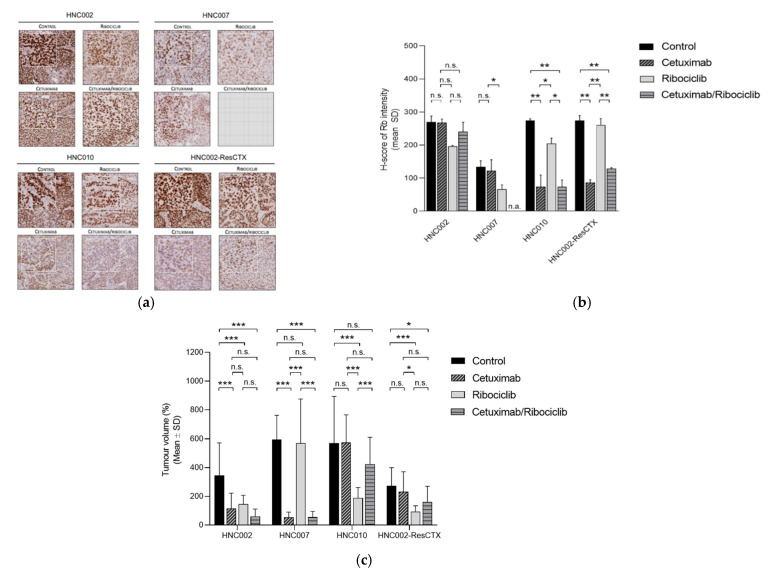
Rb protein expression levels in four HPV-negative SCCHN PDTX models (HNC002, HNC007, HNC010, HNC002-ResCTX); (**a**) Rb protein expression evaluated by immunohistochemistry (we show one representative mouse per model), and (**b**) Rb protein expression measured by histoscore in three mice/treatment/model, except for HNC002 where 2 mice/treatment were measured. In the UCLHN01 and HNC007 models, due to the important efficacy of cetuximab and/or the combination treatment, some of the tumors were too small to be analyzed for protein expression. (**c**) Tumor volumes after 42 days of treatment are represented for each treatment group, except for the tumor volume of HNC007 treated with ribociclib, which was reported after 28 days of treatment due to important tumor size. n.a.; not available, n.s. (no significance): *p* > 0.05; * *p* < 0.05; ** *p* < 0.001; *** *p* < 0.0001.

**Figure 3 cancers-13-01251-f003:**
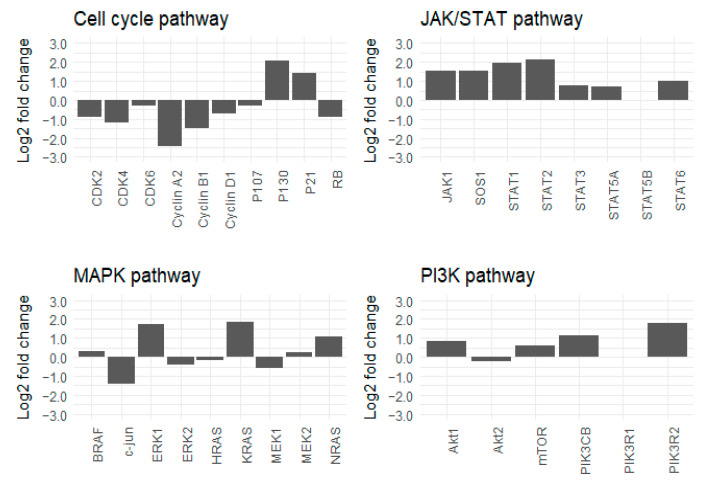
CAL27 cell line: Log2-fold change in values determined by proteomic analyses for proteins identified as important regulators of the cell cycle, JAK/STAT, PI3K/Akt and MAPK pathways.

**Figure 4 cancers-13-01251-f004:**
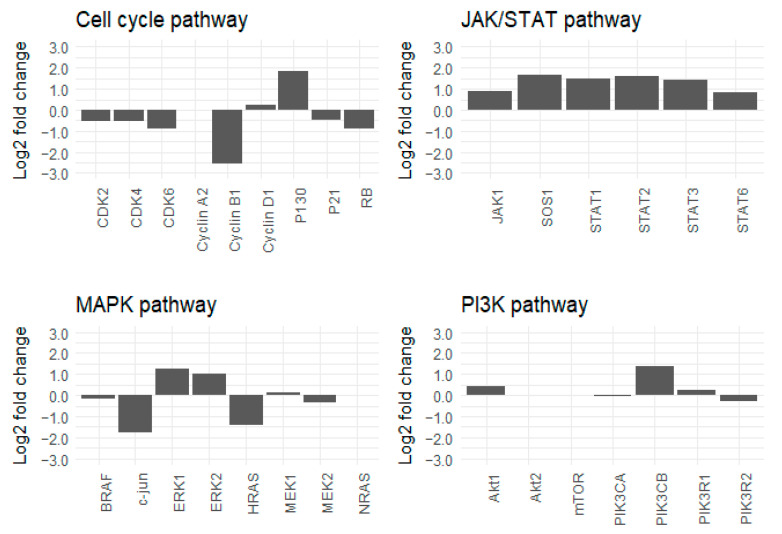
FaDu cell line: Log2-fold change in values determined by proteomic analyses for proteins identified as important regulators of the cell cycle, JAK/STAT, PI3K/Akt, and MAPK pathways.

**Figure 5 cancers-13-01251-f005:**
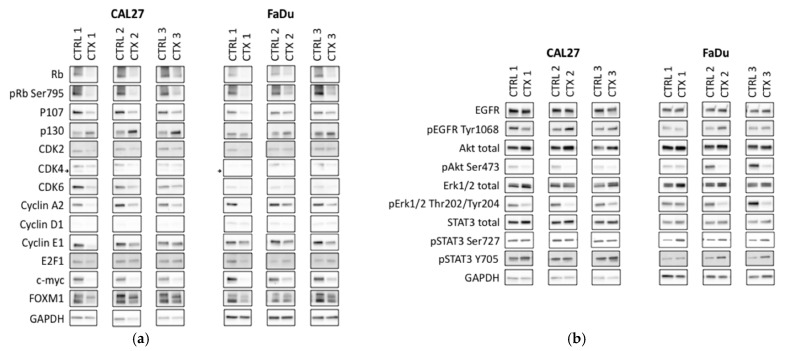
Protein expression levels of the CAL27 and FaDu cell lines. The expression of proteins associated with (**a**) the cell cycle pathway and (**b**) the PI3K/Akt, MAPK and JAK/STAT pathways was investigated in triplicate by immunoblotting. Parental and cetuximab-resistant lines were analyzed as pairs.

**Figure 6 cancers-13-01251-f006:**
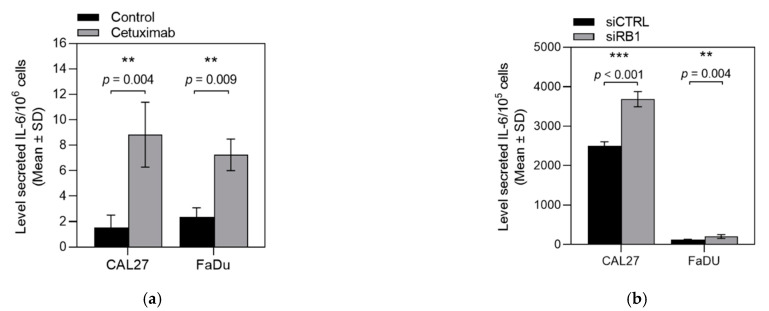
(**a**) IL-6 levels in the cell culture supernatants of the CAL27 and FaDu cetuximab-resistant cell lines compared to parental cells. (**b**) IL-6 levels in the cell culture supernatants of CAL27 and FaDu cells treated with siRNA targeting RB1. ** *p* < 0.001; *** *p* < 0.0001.

**Table 1 cancers-13-01251-t001:** *p*-Values of Rb protein expression in PDTX tumors chronically exposed to cetuximab compared to controls or ribociclib monotherapy (associated with Figure 2). In the UCLHN01 and HNC007 models, due to the efficacy of the cetuximab and/or combination treatments, tumors were too small to be analyzed for protein expression.

	HNC002	HNC007	HNC010	HNC002-ResCTX
	*p*-Values			
Control vs. cetuximab	0.999	0.707	9.50 × 10^−3^	1.60 × 10^−3^
Control vs. cetuximab/ribociclib	0.597	/ ^1^	2.30 × 10^−3^	3.20 × 10^−3^
Ribociclib vs. cetuximab	0.053	0.050	1.25 × 10^−2^	7.70 × 10^−3^
Ribociclib vs. cetuximab/ribociclib	0.257	/ ^1^	1.22 × 10^−2^	5.30 × 10^−3^

^1^ Due to the efficacy of cetuximab combined with ribociclib, tumors were too small to be analyzed for protein expression.

## Data Availability

Data is contained within the article or Appendix A. The Appendix A presented in this study are available in https://zenodo.org/badge/DOI/10.5281/zenodo.4521221.svg at doi: 10.5281/zenodo.4521221.

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
