# Peer review of "Preclinical Evaluation of the Association of the Cyclin-Dependent Kinase 4/6 Inhibitor, Ribociclib, and Cetuximab in Squamous Cell Carcinoma of the Head and Neck"

_cancers, 2021, doi:10.3390/cancers13061251_

Round 1

Reviewer 1 Report

The authors present data on their efforts to improve the treatment of squamous cell carcinoma by a combination of Cetuximab and Ribociclib. This is an interesting approach but the manuscript would improve considerably if the data presented in figure 1 and 2 would comprise the same models and chosen. Additionally, further analysis of the signaling pathways could have be done with the models, it is not clearly understandable, why for these experiments cell lines were chosen. Their approach can be improved since it does not look like that Cetuximab growth inhibition is associated with loss of Rb (HNC10 in fig1 and 2). Therefore, a more detailed approach to understand the effect might be necessary.

Reviewer 2 Report

In this work, the authors assessed the anticancer effects of ribociclib and cetuximab, alone or in combination, in HPV-negative SCCHN PDTX models. Interestingly, they showed that combined ribociclib/cetuximab was less efficacious in inhibiting tumor growth than ribociclib alone in models that exhibited resistance to cetuximab. These cetuximab-resistant SCCHN cells  also showed downregulation of Rb protein upon cetuximab treatment. Proteomic analyses followed by validation with Western blots further revealed deregulation of proteins in cell cycle, JAK-STAT, MAPK and PI3K pathways, in addition to Rb, in cetuximab-resistant SCCHN cells. Cetuximab-resistant SCCHN cells or Rb1-knocked down cells also secreted more IL-6 that has been implicated in cetuximab resistance. The authors concluded that combination of cetuximab with CDK4/6 inhibitor may reduce the activity of CDK4/6 inhibitor in cetuximab-resistant SCCHN cells. In general, this is a well-conducted study with significant clinical implications. Nevertheless, the following issues have to be addressed before their work can be further considered for publication in Cancers.

Major concerns:

  1. The authors should determine if their observed phenomenon could be generalized to other CDK4/6 inhibitors (ie. palbociclib, abemaciclib). At least, the authors should show if palbociclib/abemaciclib combined with cetuximab are less efficacious than palbociclib/abemaciclib alone in cetuximab-resistant SCCHN models.
  2. The authors should show if knockdown of Rb1 is sufficient to reduce the sensitivity of SCCHN cells to cetuximab.

Round 2

Reviewer 1 Report

The authors have improved the manuscript considerably. It is now suitable for acceptance after a language/typo check.

Author Response

The authors would like to thank again the reviewer for the careful review of our manuscript and for providing us with his/her comments and suggestions to improve its quality.

Reviewer 2 Report

I have no further comments.

Author Response

(The authors gave the same response as above.)
